# Rural Family Caregiving: A Closer Look at the Impacts of Health, Care Work, Financial Distress, and Social Loneliness on Anxiety

**DOI:** 10.3390/healthcare10071155

**Published:** 2022-06-21

**Authors:** Tanya L’Heureux, Jasneet Parmar, Bonnie Dobbs, Lesley Charles, Peter George J. Tian, Lori-Ann Sacrey, Sharon Anderson

**Affiliations:** 1Division of Care of the Elderly, Department of Family Medicine, University of Alberta, Edmonton, AB T5G 2T4, Canada; tanyarlheureux@gmail.com (T.L.); jasneet.parmar@albertahealthservices.ca (J.P.); bdobbs@ualberta.ca (B.D.); lesley.charles@albertahealthservices.ca (L.C.); peter.tian@ualberta.ca (P.G.J.T.); 2School of Kinesiology and Health Studies, Queen’s University, Kingston, ON K7L 3N6, Canada; 3Alberta Health Services, Edmonton, AB T5G 0B7, Canada; 4Medically At-Risk Driver Centre, University of Alberta, Edmonton, AB T5G 2T4, Canada; 5Department of Pediatrics, University of Alberta, Edmonton, AB T5G 2T4, Canada; sacrey@ualberta.ca

**Keywords:** rural, family caregivers, mixed methods, survey

## Abstract

Even before the COVID-19 pandemic, earlier acute care patient discharges, restricted admissions to long-term care, and reduced home care services increased the amount and complexity of family caregivers’ care work. However, much less is known about rural caregivers’ experiences. Thus, our aim in this sequential mixed-methods study was to understand how COVID-19 affected rural family caregivers. Thematically analyzed interviews and linear regression on survey data were used to understand family caregiver stress. Fourteen rural caregivers participated in interviews. They acknowledged that they benefitted from the circle of support in rural communities; however, they all reported having to cope with fewer healthcare and social services. 126 rural caregivers participated in the online survey. About a third (31%) of these caregivers had moderate frailty, indicating that they could benefit from support to improve their health. In linear regression, frailty, social loneliness, financial hardship, and younger age were associated with caregiver anxiety. Contrary to the qualitative reports that people in rural communities are supportive, over two-thirds of the rural caregivers completing the survey were socially lonely. Rural family caregivers are vulnerable to anxiety and social loneliness due to the nature of caregiving and the lack of healthcare and social service supports in rural areas. Primary healthcare and home care teams are well-positioned to assess caregivers’ health and care situation as well as to signpost them to needed supports that are available in their areas.

## 1. Introduction

Family caregivers are critical to home care. They provide 75–90% of the care to people living in the community [1,2,3] and assist with 10–30% of the care in congregate care settings such as lodges, assisted or supportive living, and long-term care [4]. Family caregivers are particularly important in rural areas. Statistics Canada defines rural as “all territory lying outside population centres” which includes small towns, villages with less than 1000 population, as well as agricultural, undeveloped, and non-developable lands [5]. Rural settings are seen as good places to grow old because of the abundance of friendly social support [6,7]. Yet rural populations generally experience significant disparities in health and healthcare resources. Disparity implies a setting in which social, economic, or environmental disadvantages contribute to inequitable outcomes [8]. Rural older adults are significantly more likely than urban residents to have functional impairments and mental and chronic illnesses [9,10] and have fewer healthcare resources [8,11]. Services such as specialist physician services, respite, and rehabilitation may not be available in rural and remote environments [12]. There may also be fewer social and community services [8,11]. It is often necessary for rural residents to travel long distances to access both health and social services [13], yet the absence of public transportation in rural settings can make access to services difficult for non-drivers [13,14]. 

Rurality also has an impact on family caregivers. While rates of loneliness and social isolation are higher for family caregivers than their non-caregiving counterparts [15,16], geographically-isolated rural caregivers are at higher risk [17]. Small and dispersed populations are barriers to health and social service provision [18,19]. In a study comparing rural and urban caregivers of people living with dementia, rural caregivers reported few formal healthcare supports compared to urban caregivers and greater reliance on informal supports [12]. Hence, family caregivers in rural communities are more likely to rely on family, friends, and neighbors for support. Younger family members, however, often move to urban areas for education or employment, which reduces the availability of day-to-day support [20]. 

COVID-19 exposed and exacerbated existing challenges [21]. Social distancing and isolating at home were effective at reducing the risk of COVID-19 infection; however, they increased loneliness and anxiety [22,23,24] and decreased social support [21,25]. Social support refers to emotional, informational, affirmational, or practical support provided by family, friends, and professionals and is a recognized external resource that can buffer the influence of stressors and bolster internal resources [26,27]. Premature patient discharges, restricted admissions to long-term care, and reduced home care services during COVID-19 increased the amount and complexity of family caregivers’ care work [22,23,24]. However, much less is known about rural caregivers’ experiences [6,28]. Thus, our aim in this study was to understand how COVID-19 has affected rural family caregivers. 

## 2. Methods

### 2.1. Design

This sequential mixed-methods study comprised of four components: (1) an exploratory mixed-methods online survey of the impacts of COVID-19 on family caregivers 4 months after the declaration of a pandemic (21 June to 21 July 2020), (2) interviews (1 February to 30 April 2021) with a convenience sample of rural family caregivers who participated in the 2020 online survey, and (3) a mixed-methods online survey of family caregivers after 18 months of the COVID-19 pandemic (21 June to 31 August 2021). Here we report on the interviews and quantitative data from the 2021 survey. The results of the 2020 survey are reported elsewhere [29,30]. 

We used a qualitative interpretive approach [31,32] for the interviews. Interpretive description is a well-established methodological approach that is designed to understand and then inform the contextual nature of human behavior and health/social care practice. This approach was used to (1) provide an in-depth understanding of the experiences of rural family caregivers during the first year of the pandemic, including what supports and services they needed due to the pandemic, (2) inform the survey planned for July 2021, and (3) generate results to inform clinical practice [33]. 

In the cross-sectional online 2021 survey quantitative data, we were particularly interested if extra care work and social isolation, which seemed common in the COVID-19 pandemic, were correlated with caregivers’ anxiety and poor health. Ethics approval was granted by the health research ethics board [PRO-00097996]. All data were collected with informed consent from caregivers. 

### 2.2. Setting

This study was conducted in the province of Alberta, Canada. Alberta is the fourth most populous province of Canada, with an estimated population of 4.08 million in 2021 [34]. One in four people are family caregivers. Educational and income levels are high—55 percent of Albertans aged 25 to 64 have completed a post-secondary program. The average annual income is $80,449 [34]. According to the 2021 census, 14.8% of the population are 65 years of age and older (629,220 Albertans). About 19% of Albertans live in rural areas.

### 2.3. Participants and Recruitment

Convenience sampling was used for both the interviews and the follow-up survey. Eligibility criteria included, (1) a family caregiver (carer, care partner) defined as any person who takes on a generally unpaid caring role providing emotional, physical, or practical support in response to another person’s disability, mental illness, drug or alcohol dependency, chronic condition, dementia, terminal or serious illness, frailty from ageing, or COVID-19, and (2) lives in Alberta. A rural living situation was determined by participant’s postal code. Participants were not compensated for their time. 

#### 2.3.1. Qualitative Interviews

Sixteen rural caregivers who completed a survey in July 2020 on the impacts of COVID-19 and indicated a willingness to participate in an interview about their experiences were invited by email to participate in individual interviews. All responded to the email invitation from the second author with agreement to be interviewed and were sent the information letter about the research and interviews. Two caregivers declined after the interviews were arranged, one because the care-receiver passed away and the other because the care-receiver was very ill. The interviews were conducted from February to April 2021 on Zoom or by telephone, depending on participants’ preference. Participants provided verbal consent and consent for the interview to be recorded before data collection began. Interviews were digitally recorded and averaged one hour in length (45–75 min). 

#### 2.3.2. Quantitative Survey

We emailed and mailed information about the survey, posters, and e-posters to health and community organizations who interact with family caregivers so they could invite family caregivers who lived in rural Alberta to participate in this study. We also advertised on social media (Twitter, LinkedIn, Facebook). The survey was delivered online via the secure REDCap data collection platform from 21 June to 31 August 2021. 126 participants completing the survey had rural postal codes. 

### 2.4. Data Collection

Qualitative interviews. A Ph.D-trained qualitative interviewer conducted all the interviews. A semi-structured interview guide was used to elicit participants’ views of how caregiving changed or remained the same during the pandemic, what was or might have been done to support them and the person(s) they cared for, and what they needed to support their caregiving in the future (Appendix A). 

Quantitative Survey. The open survey consisted of 30 closed questions (Likert scale/yes/no, list) and five open-ended questions. The questionnaire was reviewed by the research team and then the online version was tested by five family caregivers. Participants were informed that their responses were voluntary, and they could skip questions they did not wish to answer. The survey consisted of four main sections: (1) care work, (2) caregiver health (frailty, changes in physical and mental health, anxiety, social loneliness), (3) stressors (finances, navigation), and (4) demographics (of both caregiver and care-receiver). The full survey can be found in the Appendix A. The following measures were analyzed for this study.

**Care work.** Family caregivers, who were caring before COVID-19, were asked whether care work increased, remained stable, or decreased. We assessed the number of hours devoted to care work pre-COVID-19 with the following options ≤1 h, 2–9 h, 10 h, 11–20 h, 21–39 h and 40+ h. Based on the evidence that anxiety rises with care time over 20 h per week [35], we dichotomized care work to ≤20 and ≥21. 

**Frailty.** We used a self-report version [36] of the Clinical Frailty Scale [CFS] [37,38] to assess frailty. The CFS is validated for in-person frailty screening based on clinical judgment [38], but adaptations are allowed for self-report assessments of frailty [36]. Rockwood [38] defines frailty as the “term widely used to denote a multidimensional syndrome of loss of reserves (energy, physical ability, cognition, health) that gives rise to vulnerability”. The primary reason for congregate-care admission is the failing of the caregiver’s health. We reasoned that screening for frailty could enable health promotion and frailty care to maintain the caregiver’s own health. Nine CFS questions were asked to assess older adults’ frailty levels (See Appendix A) [36]. As recommended in other studies, we divided scores into three groups: CFS 1–3 fit; CFS 4–6 vulnerable with mild to moderate frailty; and CFS 7–9 severe frailty [39].

**Changes in health.** Caregivers were asked if their physical and mental health had changed in the last year (improved, remained the same, deteriorated). 

**Stressors.** We asked about self-rated financial difficulty (none, a few, moderate, a lot) and confidence to navigate health and community systems to access needed services (not at all, a little, neutral, somewhat, very, don’t know). We dichotomized these as no difficulty/confident and difficulty/not confident for some analysis. 

**Anxiety.** Anxiety was assessed with the six-item State Anxiety Scale [40]. It is a validated short form of the State Trait Anxiety Inventory [STAI]. The long and short forms are designed to measure the feelings of apprehension, tension, nervousness, and worry. Family caregivers were asked to respond to questions such as “I feel comfortable” or “I feel good” on a four-choice Likert scale (“not at all” to “very much”). Items 1, 3, and 6 are positively worded (absence of anxiety) and reversed scored). The final score was obtained by adding the scores for each item together and then multiplying the total score by 20/6. STAI scores range from 20–80, with higher scores indicating more severe symptoms. The 6-item versions have been found to be as reliable and valid as the original 20-item version [40,41,42]. Cronbach alphas range from 0.74 to 0.82 [40]. In this survey, Cronbach’s alpha pre-COVID-19 was 0.85, and post-COVID-19, it was 0.89. We dichotomized the STAI scores using cut-point scores of <40 to indicate no or minimal symptoms and ≥41 to indicate the presence of moderate or severe symptoms.

**Social loneliness.** The experience of social loneliness is related to the perception of the absence of a broader social network. We used the three-item DeJong–Gierveld social loneliness scale [43,44]. The three positively worded items, “There are plenty of people I can lean on in case of trouble”, “There are plenty of people I can count on completely”, and “There are enough people I feel close to” are rated on a scale of no, more or less, or yes. The no and more or less are scored as 1, socially lonely, and yes is scored as 0, not socially lonely. Scores range from 1 to 3. In a large sample from 7 countries, the 3-item DeJong–Gierveld social loneliness scale had good internal consistency, the Cronbach’s alpha was 0.85 [44]. In the current rural sample, Cronbach’s alpha coefficient was 0.89. 

### 2.5. Data Analysis

**Qualitative data.** Interviews were transcribed verbatim. The interviewer listened to the recordings and corrected mistakes and removed identifying information in the transcripts. The interviews were imported into NVIVO for ease of data management. We analyzed the data thematically, as suggested by Thorne [31,32]. Thematic analysis is a flexible qualitative method used to explore the different perspectives held by research participants; it highlights the similarities and divergences in their viewpoints and generates thematic insights [45]. 

We methodically followed Braun and Clarke’s [45,46,47,48] six stages of analysis (see Appendix A). Reflexive Thematic Analysis creates themes that highlight nuanced, contextualized patterns of meaning [47]. To become familiar with the data and to generate first impressions of meaning (stage one), two members of the research team independently read participants’ responses and made notes of their impressions in memos in NVIVO. In stage two, members of the research team worked separately to inductively generate initial open codes. In stage three, team members worked together to generate categories. Patterns within the open codes were identified and codes with similar attributes and meanings were grouped. The categories were then refined into preliminary themes (stage four) using critical questions such as “what is happening here”, and “what is being said here”, and “why”? At stage four, we discussed how the knowledge might apply in rural areas, influence family caregivers’ work, and how this knowledge might influence the questions we asked in the survey. We then reread the transcripts to name and confirm the final themes (stage five). The report was generated (stage six) and discussed at a final team meeting. We followed Thorne’s suggestions for quality by checking the transcripts for accuracy, using memos and keeping an audit trail [31,49]. 

**Survey data.** We used ***SPSS***^®^ 26.0 statistical software (***IBM***^®^, Chicago, IL, USA) to analyze the data. First, we generated descriptive statistics for all variables. Bivariate analyses by self-rated caregiver levels of anxiety were performed using chi-squared tests comparing sociodemographic, clinical, and caregiving characteristics between two groups of caregivers: those with State Anxiety Scale scores of ≥41 (indicating moderate to severe anxiety) versus ≤40 (no to low anxiety). Then we used hierarchal stepwise linear regression to assess the unique associations between caregiver anxiety and the demographic and contextual factors identified by the chi-square tests as significant. 

## 3. Results

### 3.1. Overview of the Qualitative Findings

In total, 14 interviews were conducted with family caregivers. Participants were all female, eight were daughters, three were wives, and four were mothers of the persons they cared for. Five were caring for congregate-care residents and nine for community-dwelling residents (See Table 1, Demographics). 

### 3.2. Qualitative Results 

There were two overarching themes (1) rural communities are a circle of support and (2) there are fewer formal supports. These rural caregivers acknowledged that they benefitted from the circle of support in rural communities, however, they all reported having to cope with fewer healthcare and social services in rural areas. This was exacerbated by the COVID-19 pandemic as indicated across three subthemes: (1) increases in care time and complexity; (2) social interaction decreased; and (3) costs increased. 

#### 3.2.1. Overarching Theme 1: Rural Communities Are a Circle of Support

Every participant referred to the support available in their rural communities through being “known in the community” and having friendly family, neighbors, and rural health and social care providers. One caregiver who also worked as a community volunteer pointed to the ad-hoc network of seniors who checked on each other, 

*So a lot of our seniors really stay connected with one another. We have our own, almost like a call-out system to check with and on each other. It is a circle of support that you need to out here*. [Caregiver to mother who lives alone in family home] 

Another caregiver to her mother in long-term care was less worried about her mother being lonely during COVID-19 lockdowns because, in their close-knit rural community, she and her mother knew the staff, 

*I feel like there are eyes and ears there. I have to confess, we’re in a bit of a unique situation here in our rural long-term care and hospital because everybody knows everybody, so I know the health care aides and LPNs and nurses that work up there. At least mom has those familiar faces. You know, whereas if we were in the city and you don’t have those connections, I don’t know how those families manage in a lockdown*. [Daughter to mother in LTC]

The mother of an immunocompromised child noted the benefits of small schools in a rural area, 


*If I were in Edmonton, I would be a lot more hesitant about the risk, but we live in a small town. It’s a huge boon to have a K to 12 school. There are only 100 kids in the entire school and they take care of each other. With the distances, there hasn’t been much spread and we keep our bubble small. I think we were pretty proactive in that.*


A wife caring for a husband recently discharged from hospital noted she was able to do so because of rural neighbors, “*She will come in if I need help, just one of those rural neighbors who are honest and just good people. That’s kind of a hallmark of people in rural areas*”. Yet one caregiver highlighted that this rural circle of support, which includes caring healthcare providers, didn’t make up for the lack of services, “*Right, it’s not a lack of caring in rural areas. Our physicians, our care teams here, care about us. They care about [husband’s name], but that doesn’t help people like us in rural areas to get services*”. 

#### 3.2.2. Overarching Theme 2: Coping with Fewer Services

According to these caregivers, COVID-19 exacerbated the existing difficulty accessing services. They reported that during the pandemic, many health providers left rural areas and the morale of those working was low, 

*I feel that the morale and understaffing are really affecting all medical personnel which was not great before COVID-19 and now is dire. They all need a grand holiday somewhere for a couple of weeks. Not practical, I know, but way more recognition than lip service and a few extra dollars that the current government is giving them. More job security and new hires*. [Caregiver to husband]

A caregiver whose husband was recently discharged after a stroke spoke about the difficulty getting home care, 

*We have never had the same services in rural areas; for instance, home care can send someone out to help you with medication in the city. They don’t give us that if you’re outside of the town. So, there are less services*. [Caregiver to husband]

A caregiver whose mother was in a lodge spoke to the lack of mental health and rehabilitation services as well as the fewer levels of congregate care. She noted that the people in her mother’s rural lodge had higher care needs than residents in urban lodges because of the limited congregate care. 

*So, from my perspective, the farther out from, you get from, the city or even some of the towns within [Name of County], it becomes difficult because we don’t have any therapists or mental health supports for these people in our communities. The rural lodges tend to have more people that need care and are less independent because we don’t have multilevel opportunities for people that can work. The lodge, that’s all we have here and if that doesn’t work, you are going to be driving a long way for a higher level of care*. [Caregiver to mother]

Caregivers said they and the Albertans they cared for needed mental health services, but they were scarce or unavailable in their rural areas, 

*You have cancelled Canadian Mental Health supports in rural, remote Alberta*. [Caregiver to husband]

*Mental supports are desperately needed now for seniors*. [Daughter caring for mother]

The mother of a child with disabilities linked mental health supports to the quality of care she could provide for her child, “*There should be mental health supports in rural areas, to make sure that parents are supported and healthy. Good mental health makes a big difference to what you can do for your child*”. Then she went on to talk about the difficulty accessing supports, “*Well, there is supposed to be all of this stuff, but I don’t even know how you go about getting it*”. 

These rural caregivers reported a lack of accessible supports and difficulty accessing supports in rural areas was exacerbated by increases in care time and complexity, reduced social interactions, and increased costs. 

#### 3.2.3. Theme 1: Increases in Care Time and Complexity

Rural caregivers who were caring for a receiver living with them were caring for a receiver living in a separate home or were designated essential family caregivers in supportive living, or long-term care noted that COVID-19 increased the complexity of the care tasks and the time they were spending providing care. Caregivers reported that earlier discharges from hospital, delayed admissions to long-term care, and difficulty accessing home care support resulted in “*more care on my shoulders*” without a break. Those caring for a person living with them noted that the time together without a break or interaction with others was taxing, 

*But the one thing I have noticed throughout this whole thing is it’s just more taxing. It’s just the constant being around each other, being home all the time. There’s no break except for respite and that kind of stuff*. [Mother of child with disabilities]

Care intensity also increased for the caregivers of congregate care residents who were one of two possible designated family caregivers, 

*It became much more intense because I was her designated caregiver and designated visitor or whatever they want to call it now. So that meant I couldn’t spread the care around at all. Everything becomes yours. So, whether it’s just visiting socially or taking her to appointments or picking up her grocery list, everything becomes your responsibility at some level. I probably doubled my hours. So, if I was going four hours a week, now I’m doing eight. It kind of depends on where she was at and her health. There was a point in time during that when she was hospitalized. I was there every day trying to visit through a window. That was very challenging*. [Daughter to mother in congregate care lodge]

Notably, while all the rural caregivers interviewed reported that time and complexity increased, those who reported that their care work had increased from ten to fifteen hours a week talked about the positive aspects of caring, such as time together, “*this time getting to know mom has been a privilege*”, and reciprocity, 

*I think and personally, I feel like. I’m doing what a good daughter is supposed to do, because I do have friends whose parents are in care in a different community and they talk to them once or twice a year on the phone. I just feel, yes closer, but yeah, that I’m doing good*. [Daughter to mother in her own home]

#### 3.2.4. Theme 2: Social Interactions Decreased

All participants elucidated on the decrease in social interactions in the COVID-19 pandemic and how the lack of social interactions and social support from others reinforced the importance of face-to-face interactions. Initially, all rural caregivers focused on the need for personally meaningful activities for the person they cared for. Seeing friends and neighbors or going to events and grocery shopping was mentioned as important to community-dwelling care receivers’ quality of life. As the quote below illustrates, several participants also commented on the impacts of the loss of social interactions with the home care staff. 

*The biggest thing is that I can no longer take them anywhere. It’s very difficult. I no longer have help to get my dad from the wheelchair into the car and out. My husband used to help, and my two siblings used to come at least once a month. They haven’t been there in over four months. Home care has pulled back a lot of things. He gets a bath once a week. They used to have a bit of housekeeping twice a month, and that’s been taken away. It was seeing someone other than me. The isolation is really difficult because my dad is going further down the dementia path because he has no external connection apart from me and it’s hard to like he’s gone down a lot*. [Caregiver to mother and father]

Those caring for congregate living residents also spoke about the importance of emotional support from family for resident’s well-being. As this quote illustrates, many caregivers thought the risk of the resident’s health deteriorating was greater than the risk of COVID-19 in these rural areas in which they had fairly few COVID-19 cases, 

*And my thing is, we cannot delay life for these old folks whose grains of sand in their hourglass are falling through at a really fast rate. They’ve got limited time. And the truth be known we’re avoiding COVID into the facility because it could kill them. You know what? It could. On the other hand, many things could kill them; mostly, they are dying from not seeing their family*. [Daughter to father in LTC]

Only after they spoke about the care-receiver’s well-being did these rural caregivers mention the impacts that care work and the COVID-19 public health protocols had on their social isolation and health. They noted that fewer resources put the onus on them to provide care and social support. One wife spoke to the isolation of working less after her husband’s early hospital discharge,

*So, I really don’t know what’s going on with the resources here. And so I’m home more, but I feel really isolated because there’s not a whole lot of resources. There’s nothing for mental health, I don’t have anybody to talk to, you know, who’s going through a similar experience, but my husband wants to stay home. When he was in hospital, he was just referred to family services for help at home. They won’t come out to our rural area. So maybe in Edmonton or bigger places, you have more resources but here, it has to be the care facility or me*. [Wife caring for husband]

Another caregiver talked about sacrificing her health to care for family in long-term care and for a child with disabilities at home while continuing with farm work, 


*Need help for myself. I’ve gained weight and become more ill caring for others. But I live in a rural area with not a lot of options. For much of my life, I feel invisible. I know there are some ways I could get more help but that feels like an extra job for which I just don’t have time or energy.*


Again, many rural caregivers submitted that public health social distancing protocols were the primary challenge that reduced social support, 

*Restrictions caused the greatest challenge....no help from others, no supports, no access to church, no access to events and activities, and no one else could help because they weren’t allowed or were too afraid to help*. [Caregiver to mother in LTC]

Caregivers did appreciate the ease of medical appointments by phone or by Internet, which they considered a boon for their physical and mental health, 

*We were delighted that we could meet with doctors and specialists by phone and by video. Doctor’s appointments are always a big deal in coordination and in his and my emotional/physical wellness. They often trigger seizures. I am exhausted. Typically, a specialist visit is an eight-hour commitment for us (preparation, travel time, waiting) for a 15 min meeting. An appointment with our family Doctor is a two-hour commitment. Telephone and video appointments for prescription updates and regular doctor and specialist appointments has been an extraordinarily positive outcome of COVID*. [Caregiver to husband] 

They also valued the social connections the Internet offered them. 

*I do Zoom calls because it’s my outlet. Yes, it’s somewhere to find resources but mainly it’s just a connection to a social network, to know that you’re not alone. You can just sit there and have a coffee or a whiskey and coke and do your thing, you know what I mean? Like just have some relaxation time*. [Caregiver to daughter with disabilities]

However, they noted that access to the Internet was expensive and unreliable in their rural areas, 

*Yeah, I’m on the Internet that goes and comes and goes, so it may disconnect for seconds. That’s the least of it, it doesn’t always keep me connected, gone in the blink of an eye*. [Caregiver to husband with stroke]

#### 3.2.5. Theme 3: Increased Costs

Lost wages and increased out-of-pocket costs were a significant concern for approximately one-third of the interviewees. Two caregivers noted incomes were reduced because they or their partner was working less during the COVID-19 pandemic. Four caregivers reported their out-of-pocket expenses increased; they mentioned increased costs in everything from the cost of insurance, taxes, electricity, heating, food, to specific caregiving needs like disability aids and incontinence products. Rural distances and lack of rural services increased their transportation costs, 

*I have to take time off from work in order to transport my dad out of town to medical appointments in the city. Sometimes these appointments are over consecutive days requiring the added expenses of hotel rooms, meals, parking and fuel*. [Family caregiver to Father]

While many participants talked about the difficulty getting home care in rural areas, one related it to transportation costs as well as home care boundaries, 

*I have been dealing with home care out of [Town 1] in the last week. They have been excellent by the way. [Town 1] is 50 km away from me making it a 100 km round trip to come out and help my husband to shower. That roughly an hour and a half to a 2-h drive round trip depending on the weather and roads. [Town 2] is 25 km away and a half-hour drive, so half the distance and time to come here but I live on the wrong side of [name of road] so home care from [Town 1] gets to come. Can the areas not work together to help each other out saving valuable time and money?* [Family caregiver to husband]

She then went on to talk about how distance made the time she had for respite inadequate, 


*I get three hours respite. Travel time is an hour, a half-hour each way, I rush doing the shopping. If I do any other errands, you rush or you can’t do it.*


### 3.3. Survey Results

#### 3.3.1. Demographics and Descriptive Statistics

In total, our recruitment strategy led to 685 link click-throughs from 21 June to 31 August 2021. Only surveys with 80% of the survey questions completed (*n* = 556) were included (completion rate 81.2%). Of the 556 Alberta caregivers participating in the survey, 126 were rural (126/556 = 22%). No cookies or IP addresses were checked to prevent multiple entries; however, we did check manually and excluded identical entries. The margin of error is not applicable in this study due to the online recruitment methodology 

Rural caregivers came from four Alberta health zones (North *n* = 39; Central *n* = 52; South *n* = 13, Calgary *n* = 22). The majority were women (89%) and between 35 to 64 (65%) years of age. About a third (31%) were over 65 and 4% were 15 to 34 years. Most (69%) had college, university, or technical training, with 15% having high school or less and 14% post-graduate or professional training. Care-receivers ranged in age from 4 to 99 years of age (Mean 66, Median 79). Two-thirds (67%) were 65 years of age and over and two thirds (66%) were living in their community homes and 34% resided in congregate care (lodges, assisted/supportive living, long-term care). Two-thirds of care receivers had one or two chronic conditions, 28% had three to four chronic conditions, and 6% had more than 5 chronic conditions. See Table 2 for demographics.

**Care work.** Most caregivers cared for one person (71%), 18% for two, and 11% cared for 3 or more. Half (38%) were caring 10 h a week or less, 15% for 11 to 20 h a week, 20% for 21 to 40 h per week, and 27% for 41 h or more per week. Caregivers caring in community homes were cared for much longer than those caring for congregate-care residents. More than two-thirds (69%) of those caring in congregate care were caring for less than 10 h a week, while 62% of those caring for Albertans in community homes were caring 21 or more hours per week. As well, 60% of the caregivers caring in community homes reported caring for more hours since the COVID-19 pandemic began compared to 33% of those caring for congregate living residents. While the proportions of the caregivers caring in community homes (85%) and congregate care (74%) providing transportation were similar, a higher proportion of the caregivers caring in community homes were helping with extended activities of daily living (78%-preparing food, housekeeping, laundry), basic activities of daily living (56% eating, dressing, managing medications), and intimate care (43%-bathing, toileting, managing incontinence). See Appendix A. 

**Caregiver health.** Over half (55%) of the respondents reported their physical health deteriorated and 68% that their mental health worsened. On the self-rated Clinical Frailty Scale (range 1–9), the mean frailty rating was 2.56 (SD1.32). Almost a third (31%) of caregivers rated themselves as moderately frail (4–6) and one caregiver self-reported as very frail (7–9). 

**Social loneliness.** The mean score was 2.12 (SD 1.23) (range 0–3). The proportion of caregivers who felt they were lacking social networks was high—72% did not have enough people they could rely on when they had problems, 71% perceived they did not have enough people they could trust completely, and 70% did not feel they had enough people they felt close to (Table 2). 

**Caregiver anxiety.** Anxiety scores ranged from 20 to 80, with a mean of 46.58 (SD 14.52). As with social loneliness, the proportion of anxious caregivers was high. Almost two-thirds (65%) of caregivers were moderately to severely anxious (scores ≥ 41). 

**Financial hardships.** Two-thirds (66%) indicated they were having financial hardship because of their caregiving responsibilities. While 22% had no increased expenses because of their caregiving, 42% were spending more on food, 27% on care supplies (incontinence products, meal replacements), 23% on medical expenses, 15% on mobility equipment, 20% on household expenses, 22% on personal items (clothing, shoes), 37% on personal protective equipment, and 27% on technology. See Appendix A.

#### 3.3.2. Regression Models

Table 3 shows the hierarchal stepwise linear regression results for the influence of weekly care time, caregiver frailty, financial difficulty, and social loneliness on anxiety while controlling for age and gender. Multicollinearity was not present (Tolerance values were more than 0.1 and variance inflation factors were less than 10). In Model 1, age and gender were entered together and explained 9.3% of the variance in anxiety. In Model 2, controlling for age and gender, weekly care time, caregiver frailty, and perceived financial difficulties improved the model fit, explaining an additional 33.5% of the variance (*F* (3, 101) = 19.72, *p* < 0.001). In the final model, four factors, frailty, social loneliness, financial hardship and age were statistically significant and explained 50.0% of the variance, with frailty recording a higher beta factor (beta = 0.322, *p* ≤ 0.001) followed by social loneliness (beta = 0.273 *p* ≤ 0.001), financial hardship (beta = 0.197, *p* = 0.016), and age (beta = −0.165, *p* = 0.032). 

## 4. Discussion

This study explored the experiences of rural caregivers who provided care during the COVID-19 pandemic. Qualitatively caregivers reported that their rural Alberta communities were supportive; however, COVID-19 exacerbated the existing lack of healthcare and social services. Care time, care complexity, and care costs increased while social interaction decreased. Quantitatively, 18 months after the World Health Organization declared that COVID-19 was a pandemic, two-thirds of the rural caregivers completing the survey were anxious, over half (55%) indicated their physical health, and two-thirds (68%) indicated their mental health had deteriorated since the COVID-19 pandemic began. Two-thirds of these rural caregivers were experiencing financial strain because of their caregiving. A plethora of research has documented the lack of supports and services for rural caregivers prior to the COVID-19 pandemic [28,50,51,52,53,54,55]. For example, in their 2006 review of rural women caregivers in Canada, Crosato and Leipart [50] found that limited access to appropriate and adequate healthcare services, challenges of distance and transportation, and social/geographical challenges left rural caregivers susceptible to distress and burden. They suggested the nurses working in rural healthcare settings were well-positioned to support family caregivers. In his 2016 book, Families Caring for an Aging America, Dr. Richard Schulz agreed that family caregivers interact with many healthcare providers in their caregiving role yet are marginalized within healthcare systems [56]. Consistent support from healthcare providers, as well as other services such as respite, community supports, and financial supports, are needed. 

In the follow-up quantitative study, weekly care work increased for a greater proportion of those caring in the community, physical health deteriorated by half and mental health by two-thirds, and almost a third of caregivers were experiencing financial hardships as a result of caregiving. Contrary to the qualitative reports that people in rural communities are supportive, in the 2021 survey, over two-thirds of the rural caregivers completing the survey were socially lonely. The three questions asked if they had enough people they could rely on when they had problems, trusted completely, or if they felt close to their networks. Family caregivers’ social networks do get smaller, especially as care time increases. In the interviews, rural family caregivers were very cognizant of the public health advice to reduce social interactions to keep care receivers safe, who are vulnerable to COVID-19 infections. 

We used linear regression to understand the factors related to the caregivers’ anxiety. Frailty, social loneliness, financial hardship, and younger age were associated with caregiver anxiety. In this survey, we used a self-report version of the Clinical Frailty Scale [36], specifically developed to explore whether person-centered care can improve outcomes once frailty and social vulnerability are identified. About a third (31%) of these caregivers reported they had moderate frailty, indicating that they could benefit from support to improve their health [37,38,57]. The primary reason for the care-receivers’ admission to congregate care is the family caregiver has died, or their health is failing [1,58]. Healthcare providers could use this self-report tool clinically to assess the family caregiver’s own view of their frailty and signpost caregivers to supports to maintain their own wellbeing. 

In this study, some rural caregivers recognized that healthcare providers in their rural areas could provide them with more support. Typically, rural healthcare providers are knowledgeable about the services and supports existing in the community and can use their knowledge of the caregiver and their care situation to signpost caregivers to person-centered supports. However, previous research indicates gap between what family caregivers need and preparation of healthcare providers to support family caregivers [2,59,60]. Even though family caregivers interact with many health providers [2,60], and caregiving advocates advise that healthcare providers are well-positioned to support family caregivers [2,56,61], current Canada health providers’ mandate is to care for their patients [59,62,63]. Health providers’ responsibility to the caregiver is not clearly delineated nor funded [59,62,63]. We agree with American caregiving scholar Richard Schultz [2] and British Columbia Doctors [61] that the family caregivers’ role needs to be recognized in policies and health providers’ role in supporting family caregivers mandated and funded. We are using this research to inform advocacy for person-centered supports for family caregivers, including mandates and funding for consistent person-centered care in the healthcare system. 

Financial burden related to caregiving was also significantly related to increased anxiety in linear regression. The relationship between anxiety and the financial burden of caregiving is consistent with prior work with caregivers [64,65,66]. In the qualitative interviews, many caregivers referred to increased out-of-pocket costs related to their caregiving during the COVID-19 pandemic that added to higher costs for transportation because of distances in rural areas. Two-thirds of these rural caregivers reported they were financially stretched due to caregiving. The rigors of caregiving can force caregivers to reduce employment, thus income, and increase out-of-pocket costs [67,68]. Using the 2012 General Social Survey, Duncan found many family caregivers were spending $500 per month, with some averaging $7000 [67]. Although there are some Canadian federal financial supports for family caregivers, very few family caregivers (6%) are able to take advantage of them [67,68]. 

### Limitations

The qualitative and quantitative data in this study are cross-sectional. Although the rural caregivers in the study came from across the province, the sample size is small. The generalizability of the findings is limited by the cross-sectional study, the small sample size, and the primarily female participants. However, rural caregivers are an understudied population. This study does demonstrate rural caregivers would benefit from support tailored for their rural area and their personal caregiving situation. 

## 5. Conclusions

Rural family caregivers are vulnerable to anxiety, financial distress, and social loneliness due to the nature of caregiving and the lack of supports, particularly evident in rural areas. Primary healthcare and home care teams are well-positioned to assess caregivers’ health and care situation as well as to connect caregivers to supports available in their areas and advocate for caregiver policies. 

## Figures and Tables

**Table 1 healthcare-10-01155-t001:** Demographics qualitative interviews.

Caregiver	Person Cared for
1. Daughter	Mother lives in a lodge
2. Daughter	Mother lives independently alone in family home
3. Daughter	Mother resides in long-term care
4. Daughter	Mother lives independently alone in a condo
5. Daughter	Mother resides in long-term care
6. Wife	Husband in their community home
7. Daughter	Mother resides in long-term care
8. Daughter	Father resides in long-term care
9. Mother	Child immunocompromised resides in their community home
10. Wife	Husband resides in their community home
11. Mother	Child with disabilities resides in their community home
12. Wife & Mother	Husband and two children with disabilities residing in their community home
13. Mother	Adult daughter, disabled since birth living in the community
14. Daughter	Mother and father living in their own home

**Table 2 healthcare-10-01155-t002:** Survey demographics by high and low anxiety.

Caregiver Factors					
Caregiver Factors		Total	Anxiety Low < 40	Anxiety Moderate–High 41+	Chi-Square *p*
Sex	Total	114	39 (34%)	75 (66%)	0.30
	Female	103 (90%)	34 (33%)	69 (67%)	
	Male	11 (10%)	5 (46%)	6 (54%)	
Age		114	39 (34%)	75 (66%)	0.39
	15–64 y	79 (69%)	25 (64%)	54 (72%)	
	≥65 y	35 (31%)	14 (40%)	21 (28%)	
Education	Total	114	39 (34%)	75 (66%)	0.58
	High school or less	17 (15%)	6 (35%)	11 (65%)	
	College University, Technical	79 (63%)	25 (32%)	54 (68%)	
	Post-graduate	18 (16%)	8 (44%)	10 (56%)	
Weekly Care time		112	39 (35%)	73 (65%)	
	Less than 10 h wk	46 (41%)	22 (56%)	24 (33%)	
	11 to 20 h wk	14 (12%)	3 (8%)	11 (15%)	
	21–40 h wk	20 (18%)	7 (18%)	13 (18%)	
	41–120 h wk	21 (19%)	5 (13%)	16 (23%)	
	121–168 h wk	11 (10%)	2 (5%)	9 (12%)	
Weekly Care time dichotomized		112	39 (35%)	73 (65%)	0.10
	≤20 h	60 (54%)	25 (42%)	35 (58%)	
	≥21 h	52 (46%)	14 (27%)	38 (73%)	
Changes in care since COVID-19 (only those who were caring before March 2020.		110	38 (35%)	72 (65%)	0.03
	More care	54 (49%)	12 (31%)	42 (58%)	
	Same care	35 (32%)	17 (45%)	18 (25%)	
	Less care	21 (19%)	9 (24%)	12 (17%)	
Length of caregiving		119			
	Mean (SD)	8.2 (7.57)			
	Median	5.0			
	Mode	3.0			
Social Loneliness					
There are few people I can rely on when I have problems		112	38 (34%)	74 (66%)	<0.001
	people to rely on	31 (28%)	20 (53%)	11 (15%)	
	Few people rely on	81 (72%)	18 (47%)	63 (85%)	
There are few people I can trust completely		114	39 (34%)	75 (66%)	<0.001
	Enough people I trust	33 (29%)	21 (54%)	12 (16%)	
	Few people I trust	81 (71%)	18 (46%)	63 (84%)	
There are not enough people I feel close to		114	39 (34%)	75 (66%)	<0.001
	Enough people I feel close to	39 (34%)	25 (64%)	14 (19%)	
	Not enough people I am close to	75 (66%)	14 (36%)	61 (81%)	
Physical Health Deteriorated		113	39 (35%)	74 (65%)	0.01
	Deteriorated	62 (55%)	15 (39%)	47 (64%)	
	Same Improved	51 (45%)	24 (51%)	27 (36%)	
Mental Health Deteriorated		114	39 (34%)	75 (66%)	<0.001
	Deteriorated	77 (68%)	13 (33%)	64 (85%)	
	Same Improved	37 (32%)	26 (67%)	11 (15%)	
Frailty Caregivers		111	38 (34%)	73 (66%)	<0.001
	Not Frail 1–3	76 (68%)	36 (95%)	40 (55%)	
	Frail 4–7	35 (32%)	2 (5%)	32 (44%)	
	Very Frail	1 (1%)		1 (1%)	
Financial difficulties		114	39 (34%)	75 (66%)	0.01
	No difficulty	39 (34%)	26 (67%)	31 (41%)	
	Difficulty	75 (66%)	13 (33%)	44 (59%)	
Ability to navigate the systems		113	38 (34%)	75 (66%)	0.23
	Not confident	22 (19%)	5 (23%)	17 (77%)	
	Confident	91 (81%)	33 (36%)	58 (64%)	
	Confident	98 (88%)	37 (38%)	61 (62%)	
**Care-Receiver Factors**					
**Care-Receiver Factors**		**Total**	**Anxiety Low < 40**	**Anxiety Moderate–High 41+**	**Chi-Square *p***
Residence		112	39 (35%)	73 (65%)	0.08
	Community	75 (67%)	22 (29%)	53 (71%)	
	Congregate Care	37 (33%)	17 (46%)	20 (54%)	
Age		108	37 (34%)	71 (66%)	0.06
	≤24 y	14 (13%)	1 (7%)	13 (93%)	
	25–64 y	21 (19%)	6 (29%)	15 (71%)	
	65–84 y	42 (39%)	15 (38%)	27 (64%)	
	≥85 y	31 (29%)	15 (48%)	16 (52%)	
Number of health conditions		107	37 (35%)	70 (65%)	0.14
	1–2	70 (65%)	26 (37%)	44 (63%)	
	3–4	30 (37%)	11 (37%)	19 (63%)	
	≥5	7 (7%)	0 (00%)	7 (100%)	
Frailty		109	38 (35%)	71 (65%)	0.89
	Active 1–3	16 (15%)	6 (38%)	10 (62%)	
	Frail 4–6	38 (35%)	14 (37%)	24 (63%)	
	Severely Frail 7–9	55 (18%)	18 (33%)	37 (67%)	

**Table 3 healthcare-10-01155-t003:** Hierarchical linear regression model for anxiety ^a^ (*n* = 111).

	Beta	95.0% CI	*p*-Value
**Model 1**			**0.006**
Gender	−0.129	−15.19, 2.97	
Age	−0.244	−4.98, −0.59	**0.013**
**Model 2**			**<0.001**
Gender	−0.010	−8.00, 7.05	0.900
Age	−0.148	−3.49, 0.13	**0.038**
Care Time	0.150	−0.172, 3.17	0.078
Frailty	0.404	2.86, 6.95	**<0.001**
Financial hardship	0.227	0.867, 5.68	**0.009**
**Model 3**			**0.001**
Gender	−0.010	−7.64, 6.65	0.891
Age	−0.165	−3.61, −0.16	**0.032**
Care Time	0.102	−0.581, 2.61	0.210
Frailty	3.853	1.89, 5.92	**<0.001**
Financial hardship	0.197	0.545, 5.23	**0.016**
Navigation confidence	−0.076	−3.27, 1.05	0.311
Social loneliness	0.273	1.37, 4.90	**<0.001**

Bolded values indicate significance at the *p* < 0.05 level. ^a^ Dependent Variable: State Anxiety Scale: 20 to 80; Adjusted R^2^ Model 1 = 0.076; Model 2 = 0.400; Model 3 = 0.465, ΔR^2^ Model 1 R^2^ = 0.093 (*p* = 0.006); Model 2 R^2^ = 0.428 (*p* < 0.001); Model 3 R^2^ = 0.500 (*p* = 0.001).

## Data Availability

Data belong to the survey Impact of COVID-19 on Family Caregivers in Alberta and the data are owned by Jasneet Parmar the principal investigator. People interested in obtaining the data set can contact Sharon Anderson, research coordinator of the study. Email sdanders@ualberta.ca.

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
