# Peer review of "Rural Family Caregiving: A Closer Look at the Impacts of Health, Care Work, Financial Distress, and Social Loneliness on Anxiety"

_healthcare, 2022, doi:10.3390/healthcare10071155_

Round 1

Reviewer 1 Report

Your study provides excellent insight into the problems faced by patients and caregivers in rural settings.

Strengths: The introduction does an excellent job of introducing the subject of the study and defines why this group of people are the part of this study. The team identified a gap in existing research, a gap where in the family care-giver was being overlooked.  This group should be looked at because they can have a direct impact on those that they care for, but more importantly the amount of care given has a direct impact on the care-giver Understanding all of this as the impetus of the study, was made clear.  The authors provided an excellent discussion of the results and provided examples fort help the reader understand .   Weaknesses:  The only weakness this document has might be the  methods section under “Design”which  needs to be a bit better spelled out. The authors need to do a better job explaining what they mean when they say sequential mixed-methods. Additionally, before now, they need to make sure they have defined what they mean by ‘rural’. That word may or may not have different connotations to people depending on where they are from.

Author Response

Reply to Reviewer 1

Thank you for this complementary review and your wise suggestions.

Weaknesses:  The only weakness this document has might be the  methods section under “Design” which  needs to be a bit better spelled out. The authors need to do a better job explaining what they mean when they say sequential mixed-methods. Additionally, before now, they need to make sure they have defined what they mean by ‘rural’. That word may or may not have different connotations to people depending on where they are from.

  1. We have rewritten the first paragraph in the methods section. Page 2 Lines 72 to 93 to clarify that the sequence was 2020 online survey, Interviews (February to April 2021), and 2021 online survey.
  2. Thank you! We should have defined rural. It is on line 37 to 38 in the introduction.

We really appreciate the time that it takes to do these reviews.   

Reviewer 2 Report

This article incorporates a sequential mixed-methods study to understand how COVID-19 affected rural family caregivers and if frailty, social loneliness, financial hardship, and younger age were associated with caregiver anxiety.  Authors assert a need for this study due to the fact that rural populations generally experience significant health and resource disparities, with rural older adults having fewer healthcare resources.  Authors further contend that rurality has impacted family caregivers, with COVID-19 exacerbating existing challenges.  To address a gap in the literature, authors aim to understand rural caregivers’ experiences during COVID-19, since much less is known about this topic.

The review is as follows:

  1. Line 72 – In “We used a qualitative interpretive descriptive approach”, a brief definition or explanation as to what is meant by a ‘qualitative interpretive descriptive approach’ would be helpful. While the section ‘2.3.1. Qualitative interviews’ explains the qualitative interview process, an initial definition of what this process is would be helpful.
  2. Line 78 – In “The interview results helped to inform the design of the 2021 survey”, specify which interview is being referred to. Is it the online survey that is being referred to?
  3. Line 83 – Consider if there is additional demographic information (e.g., income, education, occupation) to provide insight into the Province of Alberta, Canada.
  4. For section ‘2.3. Participants and Recruitment’, expand discussion on the recruitment process. Were participants compensated for their time? Line 119 mentions that the open survey consisted of 30 closed questions, so it invites the question on if participants were compensated for this time.
  5. Line 102 – The word ‘Zoom’ is written in all capital letters (i.e., ZOOM). Please review.
  6. Line 109-110 – In “We also advertised on social media (Twitter, Linked-in, Facebook)”, this information should come earlier in the manuscript. For instance, this information could be discussed in the ‘2.3. Participants and Recruitment’ Section.
  7. There is good discussion of the analysis of the qualitative and survey data.
  8. The participant quotes are insightful and compelling.
  9. Line 443 – For ‘Table 2. Survey demographics by high and low anxiety.’, it would be helpful to see the row headers repeat on the ensuing pages.
  10. Line 505 – Check spelling in ‘COVID19’.
  11. Lines 572-573 – In “Primary healthcare and home care teams are well positioned to assess caregivers’ health and care situation”, expand on this discussion. What makes primary healthcare and home care teams well positioned to assess caregivers’ health and care situation? What are future directions for this research?

Overall, this is an insightful, pertinent on a very important topic.  The study topic is unique and interesting.  Attending to some clarifying items, including the Methods and Conclusion, may help to improve the paper.

Author Response

Reply to Reviewer 2.

Thank you for the acknowledgement of the work put into this paper. We really appreciate it. Please find our reply in the attachment

Reviewer 3 Report

I congratulate the Authors for this well shaped paper. I have just a few minor comments:

Abstract

Line 29. Neededsupportsthat. Spaces missed.

Methods

It appears to me that Authors employed a quantitative – qualitative - quantitative design. Procedures for interviews and quantitative survey in 2021 are well described while it is not clear how they proceeded for the survey in July 2020. More details should be provided (e.g., How many invitation links for the survey were sent? How many family carers completed the survey?)

Line 131. I was wondering if Authors measured anxiety or distress. These are different concepts. Perhaps, the latter rather than the former is linked to care time. Moreover, anxiety has been already assessed by the STAI. Please, revise.

Perhaps some supplementary material is missed. For example, Table 1 is labelled “Types of support provided by caregivers in the community and in congregate care”, not “Stages of Thematic Analysis” as declared in the main manuscript.

Results

Some words/sentences are highlighted in yellow, remove.

Discussion

Line 498. Typo (dueing instead of during)

Author Response

Reply to Reviewer 3.

Thank you for the acknowledgement of the work put into this paper. We really appreciate it.

  1. Abstract Line 29. Neededsupportsthat. Spaces missed.

Thank you! Corrected. Not sure why that happened.

  1. Methods It appears to me that Authors employed a quantitative – qualitative - quantitative design. Procedures for interviews and quantitative survey in 2021 are well described while it is not clear how they proceeded for the survey in July 2020. More details should be provided (e.g., How many invitation links for the survey were sent? How many family carers completed the survey?)

Thank you. Every single reviewer commented that the mixed methods design needed to be clarified. We have rewritten the first paragraph in the methods section. Page 2 Lines 72 to 93 now clarifies that the sequence was 2020 online survey, Interviews (February to April 2021), and 2021 online survey. We are not reporting on the 2020 survey in this study as it is published.

  1. Line 131. I was wondering if Authors measured anxiety or distress. These are different concepts. Perhaps, the latter rather than the former is linked to care time. Moreover, anxiety has been already assessed by the STAI. Please, revise.

We are very aware of the difference between anxiety and distress. We specifically measured anxiety using Tluczek et al.  six-item anxiety scale based on research that anxiety typically rises as care responsibilities increase and energy is depleted. Notably, a single question on the RAI-Home Care instrument asks case managers to assess if caregivers are anxious, distressed, or depressed.  Those ratings begin to rise dramatically at 21 hours per week[1]. According to Janice Keefe, one of the authors of the Caregiver Risk Screen [2], it is much more difficult to intervene successfully when caregivers are distressed.  Because we want to be proactive, we think that by assessing anxiety we might be able to intervene with supports and avoid distress. 

  1. Perhaps some supplementary material is missed. For example, Table 1 is labelled “Types of support provided by caregivers in the community and in congregate care”, not “Stages of Thematic Analysis” as declared in the main manuscript.
  2. Results Some words/sentences are highlighted in yellow, remove.

You are absolutely right! Thank you. We have corrected the supplementary material and the labelling of supplementary materials. The highlighting is removed. 

  1. Line 498. Typo (dueing instead of during)

Thank you. No matter how many times we have read this, we have all missed it.

  1. Pauley, T.; Chang, B.W.; Wojtak, A.; Seddon, G.; Hirdes, J. Predictors of Caregiver Distress in the Community Setting Using the Home Care Version of the Resident Assessment Instrument. Professional case management 2018, 23, 60-69, doi:10.1097/NCM.0000000000000245.
  2. Guberman, N.; Keefe, J.; Fancey, P.; Nahmiash, D.; Barylak, L. Caregiver Risk Screen. 2001.

Round 2

Reviewer 2 Report

The authors have carefully responded to the suggested feedback. The revised paper is clearer and improved. There is also insightful added information on the demographics of residents in Province of Alberta, Canada.

One item of note.  Authors should review the information below:

1.       Lines 90-91 – Check spelling in “the first year of the htepandemic

Author Response

Thank you so much! I am dyslexic so I will read small grammatical errors like this over and over without noticing them. 

I really appreciated your very through review initially and this time too.  Thank you very much.